



# Toward quantifying turbulent vertical airflow and sensible heat flux in tall forest canopies using fiber-optic distributed temperature sensing

Mohammad Abdoli[1], Karl Lapo [1,2], Johann Schneider[1], Johannes Olesch[1], Christoph K. Thomas[1,2]

[1]University of Bayreuth, Bayreuth, Germany
[2]Bayreuth Center of Ecology and Environmental Research, Bayreuth, Germany

*Correspondence to*: Mohammad Abdoli (Mohammad.Abdoli@uni-bayreuth.de)

**Abstract.** The paper presents a set of Fiber Optic Distributed Temperature Sensing (FODS) experiments to expand the existing microstructure approach for horizontal turbulent wind direction by adding measurements of turbulent vertical wind speed and direction, as well as turbulent sensible heat flux.We address the observational challenge to isolate and quantify the weaker vertical turbulent motions from the much stronger mean advective flow signals. In the first part of this study, we test the ability of a cylindrical shroud to reduce the horizontal wind speed while keeping the vertical wind speed unaltered. Shroud experiments were performed using two sonic anemometers and two pressure ports in an open experimental area over short grass, with one observing system located inside the cylindrical shroud, but without any fiber-optic (FO) cables. The flow statistics were compared across shroud configurations of different shapes, colors, rigidity, and porosity. The white insect screen shroud with the rigid structure and 0.6 m diameter was identified as the most promising setup in which the correlation of flow properties between shrouded and unshrouded systems is maximized, and the RMSE was significantly lower. The optimum shroud setup reduces the horizontal wind standard deviation by 35%, has a coefficient of determination of 0.972 for vertical wind standard deviations, and a RMSE less than 0.018 ms⁻¹ when comparing the shrouded to the unshrouded setup. Spectral analysis showed a fixed ratio of spectral energy reduction in the low frequencies, e.g., > 2 s, for temperature and wind components, momentum, and sensible heat flux. Unlike low frequencies, the ratios decrease exponentially in the high frequencies, which means the shroud dampens the high-frequency eddies with a time scale < 6 s, considering both spectra and cospectra together. In the second part, the optimum shroud configuration was installed around a heated fiber-optic cable with attached microstructures in a forest to validate our findings, but the analysis revealed a failure to isolate the magnitude and sign of the vertical wind perturbations from FODS. However, concurrent observations from an unshrouded part of the FODS sensor in the weak-wind subcanopy of the forest (12-17m above ground level) yielded meaningful measurements of the vertical motions from coherent structures with distinct sweep and ejection phases. These signals allow for detecting the turbulent vertical airflow at least 60% of the time, and 71% when conditional sampling was applied. Comparison with vertical wind perturbations from sonic anemometry resulted in correlation coefficients of 0.35 and 0.36, which increased to 0.53 and 0.62 for conditional sampling. Evaluating the first direct sensible heat fluxes from FODS against those from the classic eddy covariance using sonic anemometry yielded an encouraging agreement in both magnitude and temporal variability for selected



periods. This observational achievement is an important step toward developing a FODS-based flux sensor capable of resolving heat flux continuously across spatial and temporal scales.

## 1 Introduction

Most fluids feature a general flow pattern combined with random motions called "turbulence" (CORRSIN, 1961). From a micrometeorological point of view, turbulence is the primary mechanism for mixing energy and matter in the air, and its

strength controls the coupling between the atmosphere and the earth surface (Burgers, 1948; Tennekes et al., 1972). Understanding the nature of turbulent flow is essential for many applications, such as the transfer and mixing of light, heat, water vapor, carbon dioxide, nutrients, and other substances, directly affecting humans, animals, and the plant's quality of life. Although current theories well describe the transport and turbulent mixing near the surface for sufficiently strong winds (Van de Wiel et al., 2012), under stable regimes, the atmospheric boundary layer (ABL) turbulence does not obey well-known

concepts, including the Monin-Obukhov similarity theory, the Kolmogorov spectrum, and Taylor's hypothesis of frozen turbulence (Grachev et al., 2013; Sun et al., 2012).

For weak-wind conditions, turbulence is not exclusively governed by dynamic stability but is driven by a combination of non-stationary processes, sub-mesoscale motions, local shear, and flow instabilities (Liang et al., 2014; Mahrt et al., 2013; Sun et al., 2012). Intermittent turbulence or non-stationary conditions often violate assumptions made for scaling laws and statistical

approaches used in micrometeorology, including translation of temporal scales into length scales through applying the assumption of Taylor's frozen turbulence. Also, the ergodicity assumption cannot be applied to the single-point measurements in these conditions. The ergodicity assumption states that the time and space averages converge under stationary, horizontally homogeneous conditions (Engelmann & Bernhofer, 2016). Several efforts have been made to analyze observations from networks of sensors to elucidate intermittent processes such as meandering and within-canopy flow and heat transport (Anfossi

et al., 2005; C. K. Thomas, 2011). Despite some progress in the mentioned studies, the lack of spatial information sufficiently dense to resolve the process scales has hindered advancing the stable boundary layer's current physical interpretation.

Fiber-Optic Distributed Temperature Sensing (FODS) was introduced as a powerful geophysical technique in the last few years. This technique measures the temperature and its location along a fiber-optic cable at a time resolution of a few seconds and spatial resolution of tens of centimeters up to 20 kilometers in total length. It is capable of observing atmospheric flows

under physically poorly understood sub-meso motions (Pfister et al., 2021; C. K. Thomas et al., 2012). Distributed Temperature Sensing (DTS) technique utility in atmospheric measurement is not limited to measuring the distributed temperature of the air, water, ice, snow, soil, and plant, but various applications such as measurement of soil moisture, humidity, shortwave radiation, and wind speed have been tested successfully (C. K. Thomas & Selker, 2021).

Among the most recent development of this technique, (Lapo et al., 2020a) introduced an approach using a FO cable with a

cone-shape microstructure printed on it, causing directional differences in the convective heat loss from the FO cable to the air. When heated coned fibers are placed in the main flow direction, the fibers on which the cones are aligned with the flow



cool more than those with opposing alignment, enabling computation of wind direction. The beforementioned study was conducted in idealized wind tunnel experiments. Very recently, Freundorfer et al. (2021) deployed this approach for the first time during actual field observation. They presented three different methods of calculating horizontal wind directions from the FODS measurements, indicating the potential of this approach to reveal the wind direction spatial structures at scales of meters and tens of seconds. The last two studies aimed at building a fully threedimensional, spatially resolving atmospheric flow sensor as an important step toward a spatially distributed flux sensor using the eddy covariance technique. Although measuring the horizontal wind speed and wind direction was examined successfully, the vertical wind component cannot be measured with existing approaches to complete the three-dimensional sensor functionality. Measuring the vertical component with the abovementioned approach is challenging as the magnitude of the vertical wind fluctuations is smaller than the combined signals of horizontal mean and turbulent flows and also the duration of the vertical events are too short, which leads to vanishing sensitivity in the vertical. Isolating the vertical wind signal by filtering the horizontal wind is essential to capture the signals induced by the air's vertical movement. Having truly 3D spatially resolved observations would create a possibility to compute the momentum and sensible heat fluxes independently of the fundamental assumptions of ergodicity, stationarity, and homogeneity since the observations are taken simultaneously in both time and space domains. For example, it could be used to reveal the generating mechanism of the weak and intermittent turbulence where the conventional observation assumptions fail. A 3D spatially resolved observation could also explain the long-standing problem of counter-gradient flux caused by a poor choice of height in the measurement and perturbation time scale (Fritz et al., 2021; Vickers & Thomas, 2014).

We addressed these conceptual and observational challenges by conducting a series of experiments. The first part of this study was conducted in the open experimental area of Bayreuth University's Ecological Botanical Garden (EBG). We isolated the horizontal wind speed using different cylindrical shrouds and compared the flow characteristics inside and outside of the shroud by placing a sonic anemometer inside and outside of the shroud to investigate (I) which are the optimal physical properties of the shroud (aspect ratio, shape, porosity, rigidity, and color) to diminish the horizontal wind disturbances adequately while leaving the vertical wind perturbations largely unaffected? And (II) how is the spectra of turbulent fluctuations affected by shroud in high and low frequencies compared to an unshrouded setup? In the second part of this study, the Waldstein-Weidenbrunnen forest environment is used to investigate the applicability of the fiber-optic approach to measuring vertical wind components since the coherent structures with strong vertical motions, the well-documemted sweep and ejection phases, in the subcanopy allows having a relatively stronger vertical wind speed compared to horizontal (C. Thomas & Foken, 2007). We deployed a shroud based on the results of the first part around heated coned FO cables in the forest area to see: (III) can vertical wind direction and wind speed be resolved by affixing a shroud around actively heated coned FO cables? If yes, how is the quality of the derived component based on point measurement?



## 2 Material and methods

### 2.1. Shroud experiment at Ecological Botanical Garden (EBG)

The first part of the measurements is conducted between April 2020 and June 2020 at the Ecological Botanical Garden (EBG)
at the University of Bayreuth. The experimental area was covered with short grass (5-15 cm) surrounded by mixed types of
trees with an approximate height of 15 meters from north, northeast, and southeast and a small artificial pond on the west side
(Pfister et al., 2017). Based on the synoptic weather station data close to the experiment location from 2014 to 2018, the
dominant wind direction for spring and summer is westerly and southwesterly, with a maximum wind speed of 8-12 $ms^{-1}$. Two
upside-down sonic anemometers (Model USA-1, Metek GmbH, Elmshorn, Germany) at 1.5 m height (middle of the shroud)
and two quad disk pressure ports attached to a nano resolution digital barometer (Model 745-16B, Paroscientific, Inc.) at 1 m
above ground set up; one of each sensor placed inside the cylindrical shroud (Fig. 1). The spacing between two sets of sensors
was 1.5 m to reduce the potential airflow distortion of the shrouded setup on unshrouded. White and gray insect screens with
a mesh size of approximately 0.5 and 0.1 mm, respectively, are used around one system. The supporting structures of the
shroud and sonic anemometers are oriented along the north-west direction, perpendicular to dominant wind speed, to reduce
the structure-induced systematic disturbances. A metal wire mesh with a 10 mm × 10 mm grid size is used underneath the
shroud to enhance its rigidity against distortion during stronger winds. The sonic anemometer and static pressure data recorded
at sampling frequencies of 20 Hz and a time scale of 10 minutes were used to compute sensible heat and momentum fluxes
and compare wind components inside and outside the shroud. In this part of the study, no Fiber Optic Distributed Sensing
(FODS) techniques were used, and the primary objective was to determine the best configuration of the shroud to meet the
study's objectives.

### 2.2. Shroud experiment at Waldstein forest

The second part of the study evaluated the utility of the shroud approach at the Waldstein-Weidenbrunnen long-term ecosystem
flux site between October 2020 and November 2020 as part of the LOEWE20 (Large Eddy ObsErvatory Walsstein Experiment
2020) experiment. The Waldstein is a forested site in Fichtelgebirge Mountains located within "Lehstenbach" catchment. This
forest was mainly covered by Norway spruce (Picea abies), characterized by variable tree heights and densities. Subcanopy
vegetation is moderately dense with a cumulative plant area index of 0.7 $m^2\ m^{-2}$, featuring shrubs with a height of ≤ 1 m. The
plant area index (PAI) is 5.6 ± 2.1 $m^2\ m^{-2}$ for the overstory trees and 3.5 $m^2\ m^{-2}$ for the understory. In the Waldstein site,
westerly and southeasterly wind with a wind speed range of 2 to 5 $ms^{-1}$ dominates in the overstory; in the subcanopy, it is
northerly and southwesterly with a range of 1 to 2 $ms^{-1}$ (Foken et al., 2017). The shroud was installed at the main tower around
a quartet FO array containing two pairs of parallel coned and unconed FO cables extending from the ground to the top canopy
at 34 m height (Fig. 2). The center of the shroud was placed at 4 m above ground, having the same diameter as the first part of
the study, but its lower boundary was located 3 m above ground. Two high-resolution DTS instruments (Model 5 km Ultima,
Silixa, London, UK) used to observe the continuous temperature with a spatial resolution of 0.127 m averaged over 3 seconds.





The DTS device connected to one of four 50-μm multimode bend-insensitive cores; inside a high-resistance stainless steel
sheath filled with gel (inner tube diameter = 1.06 mm, outer diameter = 1.32 mm, Model C-Tube, Solifos AG, Switzerland,
resistance = 1.8 $\Omega$ m$^{-1}$). The cones are made from polyethylene with a diameter and height of 12 mm, respectively, and 2 cm
of spacing optimized by (Lapo et al., 2020b). Two warm and cold solid-phase calibration baths with a constant temperature
developed by Thomas et al. (2022) were deployed at the beginning and end of FO arrays to be used as a calibration reference
(Fig. 2). A sensor package was set outside the shroud close to the center of the cylinder with a sonic anemometer (Model
CSAT3, Campbell Scientific Inc., Logan, UT, USA); a quad disk static pressure transducer (Model 745-16B, Paroscientific,
Inc.), and an open path infrared $CO_2/H_2O$ gas analyzer (Licor 7500, LI-COR Biosciences, USA). For comparison reasons, the
eddy covariance data with a sampling frequency of 10Hz at 36m height and the FODS data of quartet array from the turbulence
tower located approximately 70 meters away from the main tower were used. See Foken et al. (2021) for more details on
turbulence tower specification. The quartet fiber configuration at the turbulence tower is the same as the main tower extending
from the groung to 34m height.

## 2.3 Changepoint detection using Pettitt test

The deployed test was proposed by Pettitt (1979) and is a non-parametric test for evaluating single abrupt changes, which
applied to the ratio of the spectral energy in the shrouded setup to unshrouded. The change point computed as follows:

The first step is to calculate $U_K$ statistics using the following Eq. (1)

$$U_k = 2\sum_{i=0}^{n} m_i - k(n + 1) \tag{1}$$

Where $m_i$ represents the rank of the $i$th 1-D data when the values are arranged in ascending order, and $k$ takes values from *1,
2,..., n,* and in the second step, the statistical change point test defines as Eq. (2)

$$K = max_{1 \leq k < n} |U_k| \tag{2}$$

A change-point occurs in a series when $U_k$ reaches its maximum value of $K$. To test the significance of the detected change
points, the critical value ($K_a$) obtained with Eq. (3)

$$K_a = \left[ \frac{-1n\alpha(n^3 + n^2)}{6} \right]^{\frac{1}{2}} \tag{3}$$

Where $n$ represents the number of observations and $\alpha$ is the significance level determining the critical value (Zarenistanak et
al., 2014).

## 2.4 FODS calibration

DTS measurements were done in a double-ended configuration fashion using two channels of each DTS device (van de Giesen
et al., 2012). The start and end of each fiber were connected to two different channels, recording the temperature of the fiber
by alternating every 3 seconds between two channels. Each channel was saved separately and aggregated to a single dataset





with a time resolution of 6 seconds during the data post-processing. The coned fibers were heated electrically using the Heat Pulse Unit (HPU) system (Model Heat Pulse System, Silixa, London, United Kingdom) and applied a constant electric current of 4 wm$^{-1}$ to the stainless-steel sheath. Observed Stokes and anti-Stokes intensities were converted into fiber temperature using the pyfocs code by Lapo & Freundorfer (2020). This code uses the matrix inversion method using constant temperature sections wrapped around warm and cold solid-state reference baths. The user adjusts the bath's temperature to be set in differentiable cold and warm temperatures based on the ambient temperature range. The reference sections with a length of at least 1.8 m equal to 14 individual measurement points along the fiber were used. Finally, the artifact-free calibrated FODS temperature was used to further analysis in this study. FODS data usually contains artifacts related to the heat exchange between the FO cable and heat sinks or sources other than air, such as precipitation, solar radiation, holders, and support structures (Lapo et al., 2022).

**2.5 Shroud-microstructure approach for determining wind direction using actively heated coned FO cables inside a cylindrical shroud**

Actively heating the FO cables and maintaining them warmer than the atmosphere makes them subject to cooling through convective heat flux based on wind speed magnitude and turbulent kinetic energy. Cone-shape microstructures printed on FO cable cause directional differences in the convective heat loss from the FO cable to the air (shown in Fig. 3, right). When heated coned fibers are placed in the main flow direction, the fibers in which the cones are aligned with wind flow cool more than in opposite alignment (Fig. 3 left). The microstructure approach uses these directional temperature differences to determine the wind direction. This method was developed and tested in the wind tunnel with Lapo et al. (2020). Subsequently, Freundorfer et al. (2021) successfully deployed this method for measuring horizontal wind direction in the field experiment. This study used the same method to measure the wind direction, but in a vertical direction using a cylindrical shroud around coned fibers.

**2.6 Coherent structures detection using the quadrant analysis method**

Coherent structures in tree canopies are defined as spatially coherent motions generated with vertical wind shear, typically associated with strong vertical motions. Every coherent structure event can be divided into upward and downward motions called ejection and sweep phases (C. K. Thomas et al., 2017). This study uses coherent structure events to compare the FODS vertical wind signals against observation. The Quadrant analysis method, in combination with hyperbolic thresholding, was used to determine the coherent structures. This method uses a scatter plot of two variables ( as horizontal ($u'$) and vertical ($w'$) wind speed perturbation in this study) in a two-dimensional plane in which the coordinate of each flow variable determines the quadrant defined as *Q1* ($u'>0, w'>0$), *Q2* ($u'<0, w'>0$), *Q3*($u'<0, w'<0$) and *Q4* ($u'>0, w'<0$). The hyperbulic threshold also defines as $L = u'w'(\sigma_u \sigma_w)^{-1}$ where $\sigma_u$ and $\sigma_w$ are horizontal and vertical wind speed standard deviation which applied to select the events exceeding the specific hole size (C. Thomas & Foken, 2007). The 20Hz data of the sonic anemometer placed at 4 m agl at the main tower (as described in section 2.1.2) was used to detect the coherent structure. The outliers were





removed from sonic anemometer data, dropping the range of the data outer of $\pm 6\sigma$, assuming a normal distribution. Based on the flow statistic in the Waldstein forest, the perturbation time scale of 10 minutes was used to compute turbulence quantities. The quadrant analysis was done without applying coordinate rotation, but the 3D coordinate rotation was applied to compute the fluxes (Wilczak et al., 2001). Since the horizontal and vertical wind is decorrelated, the coherent structure events located in $Q2$ having a $|L| > 0.5$ were defined as ejection phase, and the events located in $Q4$ having $|L| > 0.5$ were defined as sweeps

phase.

**2.7 Distributed sensible heat flux using FODS**

The sensible heat flux from FODS ($w'T'$) is the product of instantaneous vertical wind perturbation ($w'$) and temperature perturbation ($T'$) which are defined as $w' = w - \overline{w}$ and $T' = T - \overline{T}$, w being the vertical wind speed which calculated using the regression of temperature differences between the upward and downward-pointing coned fibers and vertical wind speed of

the sonic anemometer. $\overline{w}$ is the mean of the vertical wind speed over both perturbation time scale (10 minutes) and length along the fiber (LAF). Similarly, $\overline{T}$ is the mean of air temperature observed with bare unheated fibers over both perturbation time scale (10 minutes) and LAF. We use $w'T'$ without averaging overbar since the perturbations from FODS are spatiotemporal averaged perturbations mostly affected by coherent structures, while in the case of eddy covariance fluxes, we use $\overline{w'T'}$ since the sonic anemometer perturbations have an inherent physical averaging by it's response time.

**3 Results and discussion**

This study was divided into two parts; the first concluded at Bayreuth EBG, optimizing the shroud configuration to filter horizontal wind speed while keeping vertical wind unaltered. During the second part, the most promising shroud configuration from the first part was deployed together with FODS in the Waldstein forest subcanopy to pursue the corresponding research objectives. In the following subtitles, the results of each part of the experiment are described.

**3.1 Shroud experiment over grass (EBG)**

During the shroud experiment at EBG, three experimental setups were tested to compare flow statistics between shrouded and unshrouded sensors. The shroud specifications are shown in the first column of Table 1. At first glance, all of the shroud configurations appear to significantly reduce the vertical wind standard deviation while keeping a good correspondence between $\sigma_w$ inside and outside the shroud, decreasing during the night and increasing during day times (Fig. 4a,b). Also, the

scatter plot between the shrouded and unshrouded $\sigma_w$ shown in Fig 4. d, e, and f confirm a high linear relationship (e.g., $R^2 > 0.9$) between shrouded and unshrouded $\sigma_w$ during the night and day times. The coefficient of determination ($R^2$) in setups 2 and 3 with 0.975 and 0.973 reveals that the shroud with 60 cm diameter reduces the vertical wind standard deviation less than the 1m diameter shroud. Also, setup 3 is less scattered where the 95% prediction interval is narrower compared to other setups. A prediction interval is a type of confidence that predicts the value of a new observation based on your existing model





(Neter et al., 1996). The Root Mean Square Error (RMSE) shown in Table 1 shows the lowest error of 0.012 ms$^{-1}$ for daytime and 0.017 ms$^{-1}$ for nighttime for setup 3. Note that the unrotated wind statistics are used for comparisons since the reduced horizontal wind speed inside the shroud caused unphysical results when applying coordinate rotation.

Besides vertical wind standard deviation, the scalar wind speed and horizontal wind speed added to the comparison by computing the reduction percentage, coefficient of determination ($R^2$), and root means square error (RMSE) shown in Table 1

for the three setups. As shown in the table, setups 1 and 2 reduce the scalar and horizontal wind speed by 72 and 74%, which induces a 30 and 34% reduction in the vertical wind standard deviation. While in setup 3, the reduction is 35% for scalar and horizontal wind speed and 23% for vertical wind standard deviation. In terms of coefficient of determination and RMSE, the third setup has higher $R^2$ while having lower RMSE than the other setups. However, using a 60 cm diameter dense shroud reduces horizontal and vertical wind speed (w) more than the third setup but reduces the goodness of fit between shrouded and

unshrouded wind components significantly.

Spectral analysis was performed to compare the shrouded and unshrouded instruments for the third setup to investigate the frequency-specific impacts. Fig. 5a and b show that the cylindrical shroud around a sonic anemometer reduces the energy in the integral scales (low frequencies) in both spectra and cospectra. In the inertial subrange, the energy decay slope for both shrouded and unshrouded setups remains mostly -5/3 for spectra and -7/3 for cospectra which is based on Kolmogorov(1941).

It confirms that the shroud has a minimal effect on the isotropic homogenous eddies in the inertial sub-range. The significant effect of the shroud can be seen in the high frequencies closer toward the dissipation scales, where the shrouded and unshrouded spectra and cospectra deviate from each other significantly (Fig 5 c,d). The ratios decrease exponentially for high frequencies, which means the shroud dampens the high-frequency eddies extremely. There is a fixed ratio of spectral energy reduction in the low frequencies for temperature and wind components, momentum, and sensible heat flux in both spectral densities. The

Pettit test detected the time scale of the change point at which the spectral energy decreases abruptly. The change points are shown in Fig.5 (c and d), with vertical dotted lines for each spectrum varying approximately 2 and 6 seconds. It means the eddies smaller than 6 seconds are highly influenced by the shroud and should be considered in further analysis.

The results of the shroud experiment at EBG indicated that the third shroud setup is the most promising setup having the slightest disturbance on vertical wind within the shroud while reducing the horizontal wind speed by 35%. Additionally, the

spectral analysis revealed that installing the shroud significantly dampens the high-frequency eddies (less than 6 seconds), which should be noted in future deployments in the forested environment.

**3.2 Shroud experiment in the forest (Waldstein)**

The third shroud setup was deployed during the LOEWE20 experiment around coned FO cables to evaluate if the vertical wind speed and direction could be observed using a combination of the shroud setup and heated coned fibers. In order to adjust

to the forest environment dominated by sweep ejection cycles of coherent structures with larger spatial scales, the height of the shroud was increased from 1.5 m in the first part of the study at EBG to 3 m. Fig. 6a shows a series of reddish and bluish strips of ΔT within both boxes, most likely induced by stronger updrafts and downdrafts, respectively. The depicted structures





outside the shroud are clear and more organized than within the shroud, and ~~seem to~~ reflect the vertical air movements better than inside the shroud. Fig. 6b magnifies the inside of the shroud, adding the 6s averaged vertical wind speed to give an idea

of how $\Delta T$ reacts to vertical wind speed changes in which there is no apparent correlation between the two. The correlation coefficient between $\Delta T$ and w ($\rho_w$) in the main tower is 0.024 inside the shroud but increases to 0.35 outside, demonstrating the substantial increase in correlation between $\Delta T$ and w outside the shroud, where the horizontal wind speed in the subcanopy is at its lowest. The $\rho_w$ coefficient further improved when the data subsampled based on rolling correlations between $\Delta T$ and w ($\rho_{roll}$) more than 0.8, climbing to 0.53 at the main tower outside the shroud and 0.62 at the turbulence tower. However, the

subsampling significantly improved the correlation outside the shroud for both towers, but no improvements were made inside the shroud, as shown in Fig. 7a. The $\Delta T$ within the shroud does not show a good agreement with vertical wind speed, and the results show the failure of the shroud configurations used in the forest experiment to achieve the research objectives. Potential reasons for the failure of this method can be summarized as follows: (i) The 3m long shroud might act differently than the 1.5m shroud to suppress the eddies that are passing through the shroud (ii) The percentage of the shroud's horizontal wind

reduction is not large enough to keep the horizontal wind speed below the speed observed with coned fibers inside the shroud. (iii) strong vertical motions resulting from passing coherent structures, which are expected to induce an explicit $\Delta T$ at greater heights, may slow down during penetrating from above the canopy in the sweep phases and not accelerate enough at the shroud height during the ejection phase. The results of Thomas and Foken (2007) studying flux contributions of coherent structures at this site support this argument since the sensible heat fluxes are well coupled within the canopy up to a height of 0.72h, h

being the canopy but in lower heights (e.g., 0.29h), the correlation between top and subcanopy is weak.

The analysis of the optimum shroud configuration at Waldstein forest failed to observe the sign and magnitude of the vertical wind perturbation using FODS; however, conducting this experiment coincided with an unexpected discovery that the heated coned fibers could observe the coherent structures events in the weak-wind subcanopy outside the shroud within the heights where the minimum horizontal wind speed occurs in the subcanopy. See Fig. A1 in the appendix for the distributed wind speed

profile calculated with FODS for the period of the data used in this study. Fig. 6c shows the mentioned height range together with the sweep and ejection phases, illustrating a good agreement between the positive $\Delta T$ with ejection phases and negative $\Delta T$ with the sweep phases. The red and blue arrows follow the reddish and blueish $\Delta T$'s, where higher vertical wind speeds are accompanied by solid features of reddish or blueish stripes, which indicates the duration of coherent structures, while the color intensity might represent the vertical wind speed of motions. In contrast, conditional sampling does not improve the

correlation between $\Delta T$ and w inside the shroud (Fig. 7a). Note that the sonic anemometer data at 4m height was used to compute the correlation coefficients with a section of FODS data at the main tower at 12 to 17m height, adding two sources of uncertainty to the correlation. First, there may be a time lag between two measurements because of height differences, and second, the two heights may not be well coupled, so strong vertical movements of air at 12-17m height might dissipate or weaken at 4 meters. The mentioned limitation at the main tower is expected to be less at the turbulence tower since the sonic

anemometer is placed at 36 m above the canopy level, where there is a robust sensible flux coupling between these two heights based on Thomas and Foken (2007). The scatter plot between $\Delta T$ and w shown in Fig.7b and c confirms a slight positive





correlation between ΔT and w with $\rho_w$ = 0.35 and 0.36 at the main and turbulence tower for the whole selected period shown in yellow, respectively. This correlation improves under certain situations: for instance, the correlation coefficient increases to 0.53 and 0.62 when conditional subsampling is used based on the rolling 5-window correlation of ΔT and w ($\rho_{roll}$) greater than

0.8 (shown in black in Fig. 7b and c). The best fit for the median of ensembled w over ΔT is the quadratic function for both towers with an $R^2$ of 0.78 and 0.93 and an RMSE of 0.017 and 0.14 ms$^{-1}$ for the main and turbulence tower. The quadratic relation between w and ΔT is in line with similar equations describing the relation between convective heat loss from the fiber-optic cable and horizontal wind speed, including Sayde et al. (2015) and van Ramshorst et al. (2020). The linear model is also the second best fit model to the median points of the ensemble vertical wind speeds over the ΔT with an $R^2$ of 0.70 and 0.88

and RMSE of 0.044 and 0.184 for main and turbulence towers (Table 3). The height of the error bar in Fig. 7a and b remains almost the same at different ΔT, but the scatter plots are more scattered in the low ΔT. The horizontal wind speed was found as the main reason for deteriorating the ΔT signal, where the lowest horizontal wind speed shows smaller error bars (Fig. 8a and b). However, the horizontal wind speed of less than 0.3 ms$^{-1}$ at the subcanopy includes a few percent of the data, which was insufficient to separate and analyze the relationships within these periods.

Following the comparison between ΔT and w for a selected 7-hour data for both main and turbulence towers, we evaluated the dataset for the percentage of correctly detected vertical wind direction, which is calculated as a fraction of the number of vertical wind directions that were successfully observed using FODS to the number of chosen data (F). A section of 5 m length of FODS ΔT from 12 to 17 m agl was selected and compared against sonic anemometer vertical wind direction to compute the F. Table 3 shows F for whole data and when the 5-window rolling correlation ($\rho_{roll}$) between ΔT and w is larger than 0.8. The

analyzed data shows 60 and 63% F at the main and turbulence towers, respectively, which increases to 71 and 67% for $\rho_{roll}$ > 0.8. Since we know the sign of the distributed vertical wind perturbation with a defined F, we can compute the magnitude of the distributed vertical wind speed perturbations from the relationship between vertical wind speed and ΔT with a known correlation coefficient and have distributed air temperature perturbations directly observed from the unheated FO cable across the entire canopy; we computed the first-ever distributed sensible heat fluxes ($w'T'$) solely based on FODS using the linear

and quadratic regression between w and ΔT in Fig. 7b and c to compute the vertical wind perturbation ($w'$) for the unshrouded section. The distributed heat flux in Fig.9 and 10 is based on a linear regression between w and ΔT at the turbulence and main towers, which yields an RMSE of 0.0009 and 0.01 Kms$^{-1}$ in comparison with the sonic anemometer's sensible heat flux ($\overline{w'T'}$). However, the RMSE is lower at the main tower, but there is a minor correlation of 0.035 between computed flux and observation, whereas the correlation is 0.26 at the turbulence tower. The same flux computation as Fig. 9 and 10 is computed

using quadratic regression shown in the appendix (Fig. A2, A3). Despite the many assumptions creating uncertainty in computing distributed sensible heat flux, the range and temporal dynamics of the computed fluxes across techniques match well for this selected period. We observe two areas with distinct differences (Fig. 9a and 10a): (i) the area near to ground up to about 7 m where the fluxes are negative and downward. Interestingly, the features are more pronounced between 05:00 and 06:00 when the horizontal wind is weaker, (ii) the height between 9 and 20 m, where the fluxes are often positive and upward,

and the features of updrafts are discernible most of the time since the horizontal wind speed at the mentioned height remains





below the threshold (~ 0.35 ms$^{-1}$ based on Fig. 8a). The comparison also confirms that the computed sensible heat fluxes are in the range of observation, but still, there are overestimation and underestimation at some periods (Fig. 9b and 10b). As discussed in section 2.7, the fluxes shown as $w'T'$ and $\overline{w'T'}$ in this study are inherently not identical and we don't expect them to be compared precisely. Nonetheless, it is the first-ever trial to compute a spatially distributed sensible heat flux using FODS which reinforces the microstructure approach's potential to observe vertical wind direction and wind speed with higher accuracy.

## 4 Conclusion

This study examined the potential of a physical shroud constructed around heated and unheated fiber-optic cables to overcome the limitations of the microstructure approach caused by the much stronger horizontal wind speed to observe distributed vertical wind speed and direction from FODS. We arrive at the following conclusions:

I.    Comparison between different shrouds in shape, color, rigidity, and porosity suggested that the white shroud with a 0.60 m diameter is the optimum shroud suitable for filtering the horizontal wind speed by 35% while having minimal effect on vertical wind speed.

II.   The spectral analysis between shrouded and unshrouded setup showed a significant energy reduction dissipation range. This reduction is minor in the inertial subrange, and both shrouded and unshrouded spectra obey the energy decay law. The spectra ratio between shrouded and unshrouded setup revealed that the eddies smaller than 6 seconds are extremely damped within the shroud.

III.  Validating the promising shroud setup by adding heated coned fiber inside showed that the shroud fails to keep the vertical wind signal. This failure could be because (i) the shroud doesn't reduce the horizontal wind speed below the threshold that the coned fibers sense the signal. (ii) The shroud increased length to 3m changes the behavior of the shroud against the eddies in comparison with a 1.5m shroud and causes suppression of the eddies that are dominant at the shroud height. (iii) The Shroud installation height is not dominant with the same strong coherent structures as the comparison section (12-17 m). In other words, the canopy is not fully coupled with the lower height of the canopy.

IV.   In contrast, evaluation of FODS observations from unshrouded paired coned and heated and unheated FO cables revealed that there is a height range in subcanopy (12-17 m) where the ΔT from coned fibers show good agreement to sonic anemometer observations. This agreement is more significant during the vertical air movement of coherent structures (e.g., sweep and ejection phases). Within the mentioned height range, the heated coned fiber is capable of detecting the vertical wind direction at least 60% of the time. The vertical wind speed also has a correlation of 0.35 with the temperature difference between up and down-pointing fibers, which improvers up to 0.62 with conditionally sampling.



V.     The distributed sensible heat flux was computed using the linear and quadratic regression of ΔT and w. Nonetheless, the distributed sensible heat flux seems within the plausible range of the measured fluxes with eddy covariance but still needs improvement. The spatial structure of the distributed sensible heat flux shows explicit information about flux distribution along with the canopy height and changing the sign of the flux at about 7m height. This study could be followed with further improvements by adjusting the cone shape and size to increase the correlation of ΔT and vertical wind direction and wind speed. Additionally, deploying DTS instruments with a high signal-to-noise ratio (SNR) and a FO cable with a low response time would lower the uncertainty.

*Funding.* This research was funded by the European Research Council (ERC) under the European Union's Horizon 2020 research and innovation program (Grant Agreement 724629 DarkMix)

*Author contributions.* Conceptualization, M.A., K.L., C.K.T; methodology and manufacturing, M.A., C.K.T, K.L., J.S. and J.O; field data collection, M.A., K.L., and J.S; validation, M.A.; formal analysis, M.A.; investigation, M.A., C.K.T.; data curation, M.A.; writing---original draft preparation M.A.; writing---review and editing, all authors; visualization, M.A.; project administration, C.K.T, M.A.; funding acquisition, C.K.T. All authors have read and agreed to the published version of the manuscript.

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



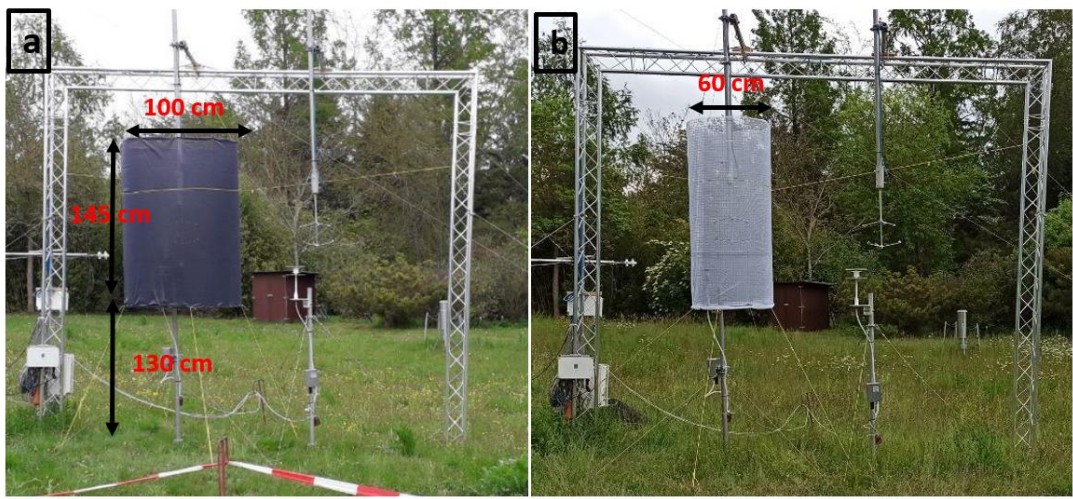

**Fig. 1. Two examples of shroud experiment setup containing two upsides down sonic anemometers and two high precision pressure**
**transducers with the quad disk to measure static pressure. (a) A 1 m diameter cylindrical shroud with fine porosity gray color texture**
**(b) A 60 cm cylindrical shroud with a white insect screen.**





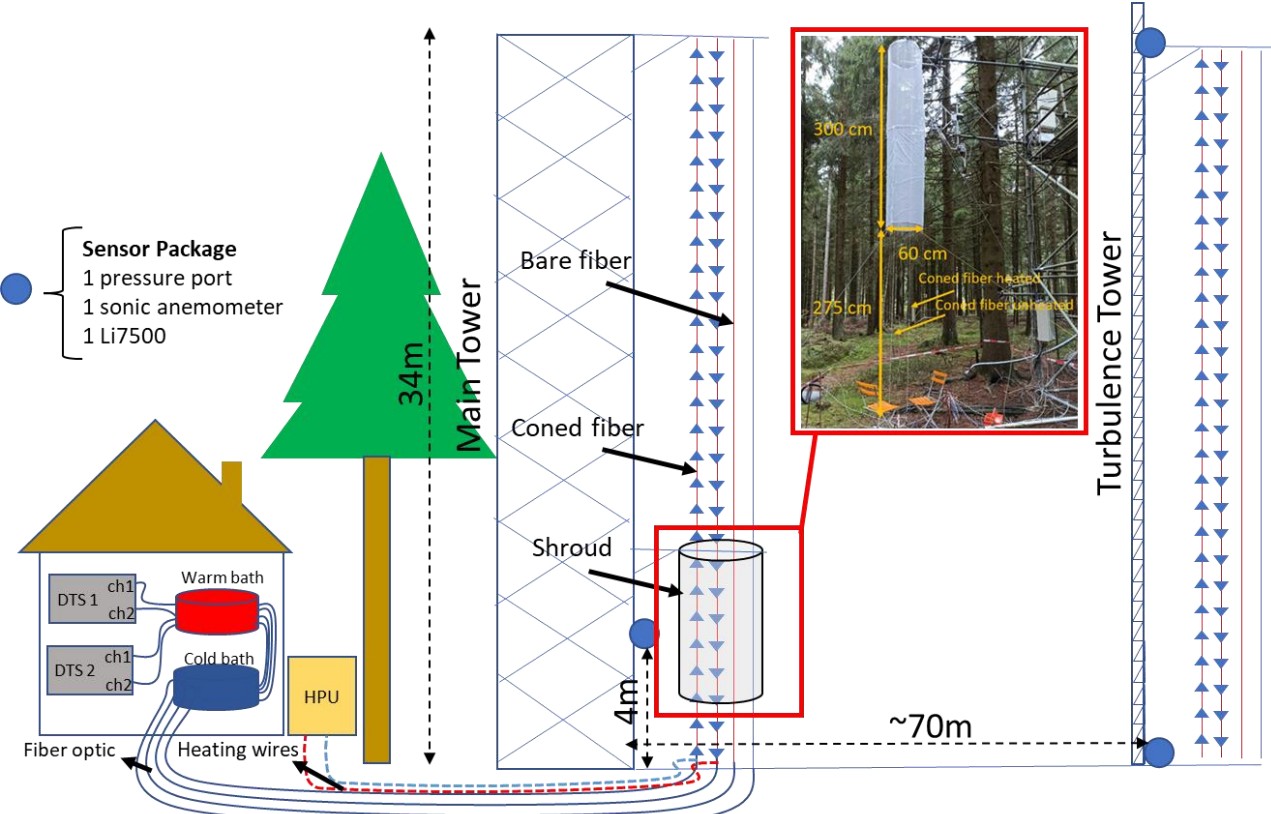


**Fig. 2. Left: Shroud experiment schematic at Waldstein forest including a quartet of a vertical FO array and affixing a shroud at 4m height at the main tower close to sensor package including a static pressure port, a sonic anemometer, and an open path infrared gas analyzer. Each coned and unconed fiber is connected to separate DTS devices. For DTS data calibration purposes, warm and cold baths were deployed at the start and end of each fiber optic loop. Three FO arrays out of four are heated using Heat Pulse Unit**
**(HPU). Right: Photograph showing the subcanopy shroud experiment and the same vertical quartet FO array at the turbulence tower having the two sensor packages as the main tower at 0.1m and 36 agl.**





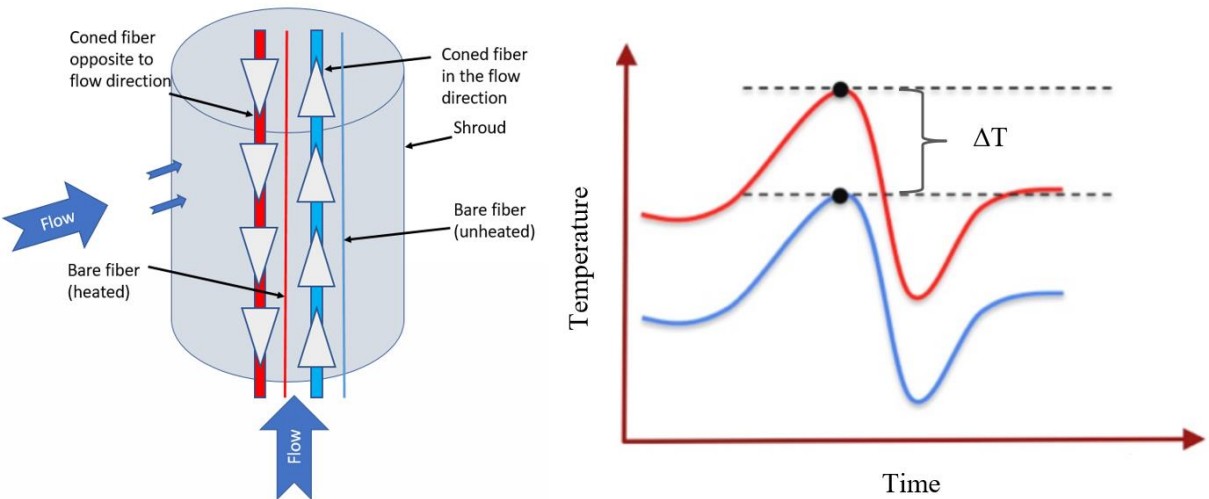

**Fig 3. Left: Microstructure approach to detecting vertical wind direction using coned FO cables inside the cylindrical shroud. Assuming updraft, the left fiber (indicated in red) would be warmer than the right fiber (shown in blue) due to the cones' direction. Right: Microscopic approach: ΔT determined by wind direction.**




**Fig 4.** Time series of unrotated vertical wind speed standard deviation ($\sigma_{w,unrot}$) for the shroud experiment at EBG are shown for
setup 1 (a), setup 2 (b), and setup 5 (c) for shrouded and unshrouded sonic anemometers. 10 min statistic of unrotated vertical wind
speed was used to compare the setups. Scatter plots between shrouded and unshrouded vertical wind speed (w) standard deviations
are shown for setup 1 (d), setup 2 (e), and setup 3 (f), separated for night and daytime, shown with black and yellow solid points.
The linear prediction and 95 prediction intervals have also been shown with blue and magenta solid lines.



**Fig 5. (a)** The spectra of the wind speed components and temperature of the third setup computed for every 10 minutes and ensemble-averaged over the whole period (5 to June 14 2020), **(b)** the cospectra of the horizontal momentum fluxes and buoyancy fluxes for the shrouded and unshrouded setups, using 20 Hz sampled data, **(c)** the ratio of shrouded to unshrouded power spectra, and **(d)** the ratio of shrouded to unshrouded cospectra specifying the change points with vertical dotted lines. The colors used in (c) and (d) for dashed lines show the same variables in (a) and (b).







**Fig 6. (a) Differences in upward-pointing and downward-pointing FODS temperature (ΔT) at the main tower during the LOEWE20**
**experiment on October 24 2020, 05:00 to 06:00 UTC. The black box with a dashed line shows the location of the shroud installation,**
**and the black box with a solid line shows the comparison section. (b) shows the magnified temperature difference inside the shroud**
**containing 2.75-5.75 m above the ground. The solid black line shows the vertical 6s wind speed average of the sonic anemometer**
**installed in the middle of the shroud. (c) shows magnified temperature within the solid black box in (a), the height range where the**
**minimum horizontal wind speed occurs in the subcanopy. The red upward arrows show the vertical wind during the ejection phase,**
**and the blue downward arrows show the vertical wind during the sweep phase extracted with quadrant analysis with hyperbolic**
**thresholding using 20 Hz sampled data at 4m height.**



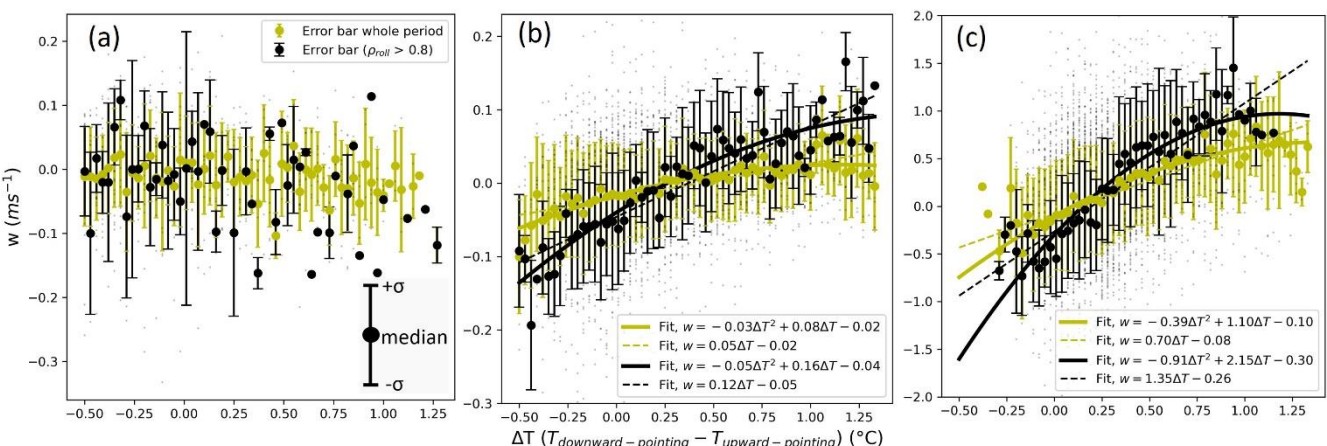

**Fig 7. (a) Scatter plot between vertical wind speed (w) of sonic anemometer at 4m height and ΔT inside the shroud (3-5 m height) for the whole period shown in yellow and subsampled with ρ$_{roll}$ >0.8 with black. The error bars show the median with a solid point with two whiskers at ±σ. (b) the same scatter plot as (a) but outside the shroud at 12-17 m agl. At the main tower. (c) The same scatter plot as (b) but for turbulence tower. The error bars are based on the ensemble average of w over ΔT with a bin size of 0.03 °C, and the fit lines are based on the median points for each bin. The data of October 24 2020, 01:00 to 08:00 UTC was used to produce the scatter plots. All the data in scatter plots are limited to the 99% percentile.**

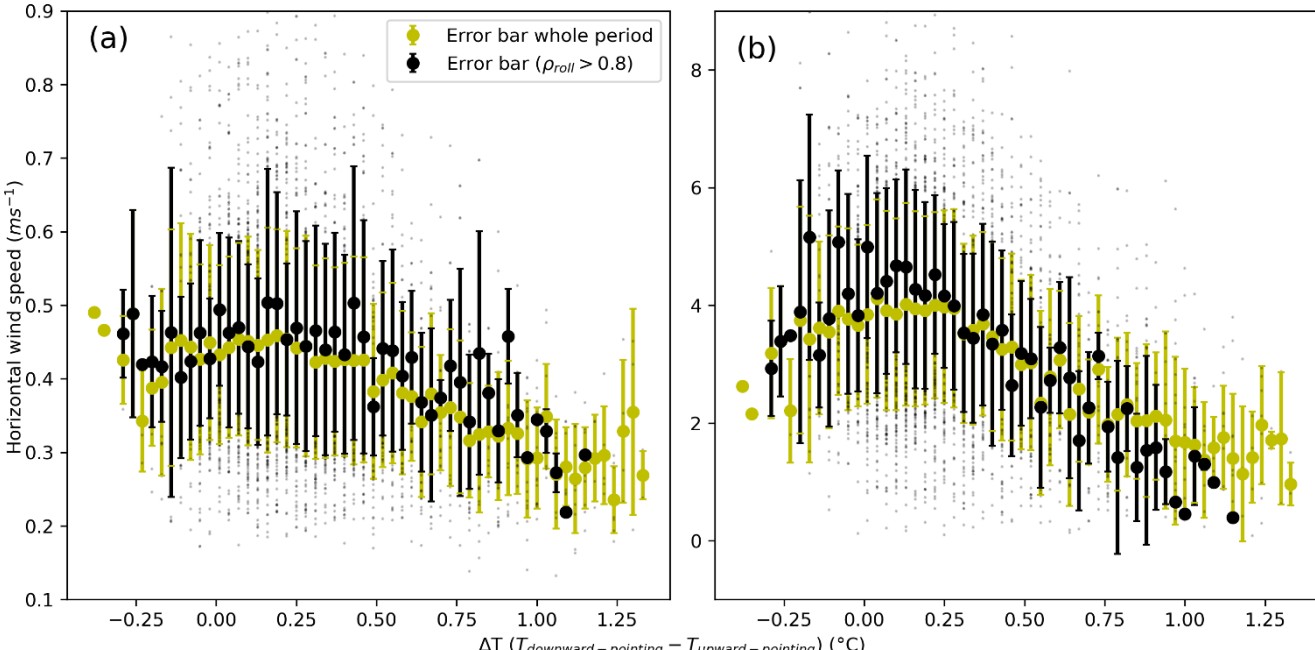

**Fig 8. (a) Scatter plot between horizontal wind speed computed using FODS and ΔT both averaged at the 12-17m agl at the turbulence tower (b) The same plot as (a) but using the wind speed from sonic anemometer at 36m agl. See the Fig. 7 caption to the error bar and data information.**

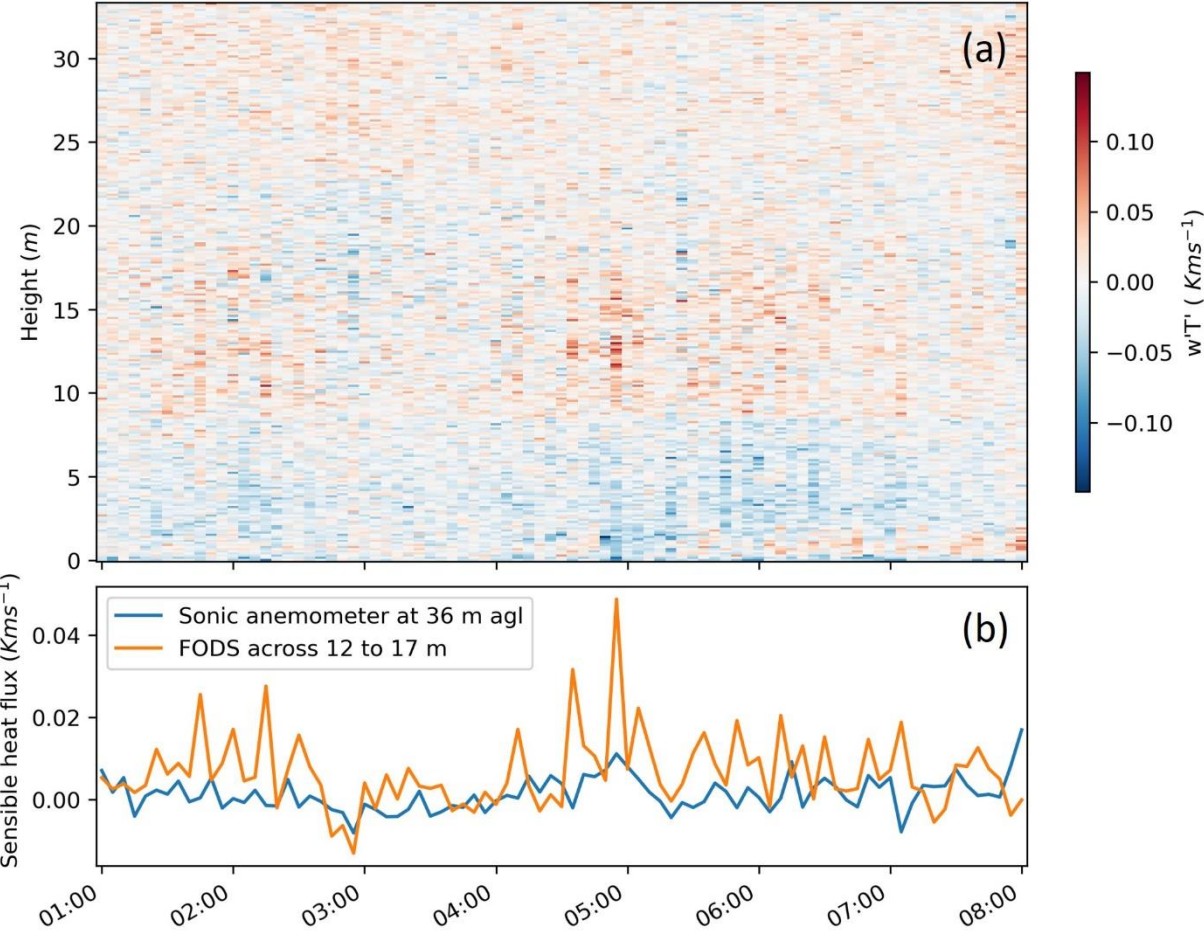

**Fig 9. (a) First-ever distributed sensible heat flux estimates from FODS averaged over 5 minutes at the turbulence tower on October 24 2020, 01:00 to 08:00 UTC. Our approach utilizes a linear regression between ΔT drove from coned fibers and w from a sonic anemometer at 36m to calculate distributed w. The temperature of unheated bare fiber is used as distributed air temperature. (b) Time series of the sensible heat fluxes observed with the sonic anemometer and from FODS for heights between 12 and 17m, which is the height where the horizontal wind speed is minimum.**




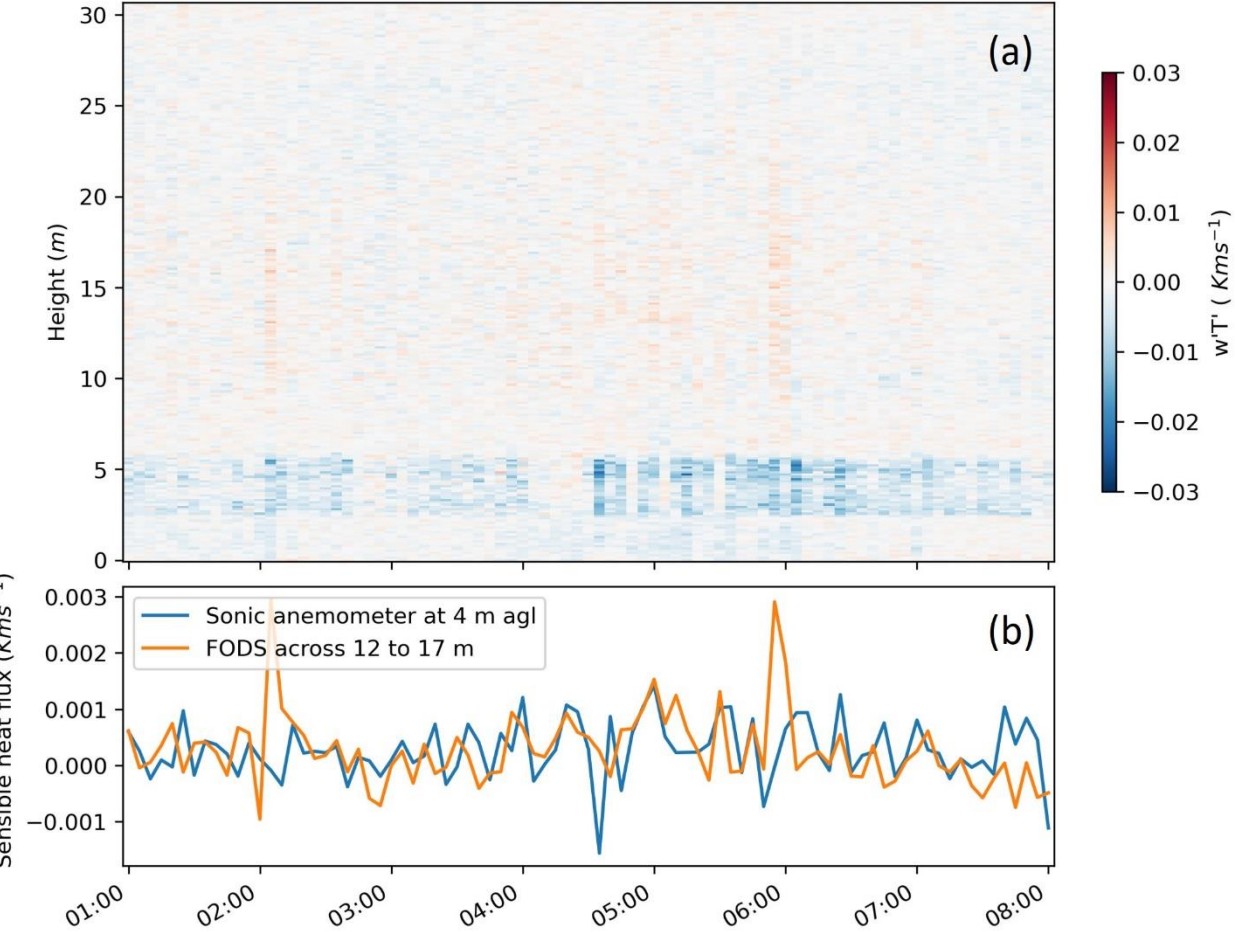

**Fig 10. (a) Distributed sensible heat flux estimates from FODS averaged over 5 minutes at the main tower using a linear regression**
**between ΔT drove from coned fibers and w from a sonic anemometer at 4m to calculate distributed w on October 24 2020, 01:00 to**
**08:00 UTC. The temperature of unheated bare fiber is used as distributed air temperature. The highlighted bluish fluxes are induced**
**within the shroud. (b) Time series of the sensible heat fluxes observed with the sonic anemometer and from FODS for heights between**
**12 and 17m.**







**Table 1. Coefficient of determination and RMSE between scalar wind speed, horizontal wind speed, and vertical wind standard deviations with and without shroud setup with N being the number of averaging intervals of 10 minutes statistic. (1-a)\*100 shows the reduction percentage of each component with the shroud. RMSE is calculated between data and the linear fit. The RMSE for each variable is shown in a range referring to the RMSE of daytime and nighttime data. The observational period is April 15 to June 15, 2020.**

| Statistics / Setups | N | Scalar wind speed | | | Horizontal wind speed | | | $\sigma_{w,unrot}$ | | |
|---|---|---|---|---|---|---|---|---|---|---|
| | | (1-a)\*100 | $R^2$ | RMSE [ms$^{-1}$] With - without | (1-a)\*100 | $R^2$ | RMSE [ms$^{-1}$] with- without | (1-a)\*100 | $R^2$ | RMSE [ms$^{-1}$] with - without |
| Setup1(Gray & dense shroud, 1 m daimeter, 1.5 m height) | 553 | 74 | 0.945 | 0.16-0.15 | 72 | 0.952 | 0.39-0.4 | 30 | 0.935 | 0.014-0.019 |
| Setup2 (Gray & dense shroud, 60 cm daimeter, 1.5 m height) | 550 | 74 | 0.946 | 0.19-0.17 | 74 | 0.955 | 0.53-0.51 | 34 | 0.975 | 0.018-0.024 |
| Setup3 (White & not dense shroud, 60 cm daimeter, 1.5 m height) | 1305 | 35 | 0.961 | 0.033-0.045 | 35 | 0.978 | 0.1-0.15 | 23 | 0.972 | 0.012-0.017 |

**Table 2. Coefficient of determination ($R^2$) and RMSE between quadratic and linear fit on the median points at Fig. 7 for the whole period and $\rho_{roll}$ >0.8. The number of the data used to compute the $R^2$ and RMSE was N = 62 for all cases. The same data as Fig.7 was used for computation.**

| | Main tower whole period | Main tower $\rho_{roll}$ >0.8 | Turbulence tower whole period | Turbulence tower $\rho_{roll}$ >0.8 |
|---|---|---|---|---|
| $R^2$, linear model | 0.70 | 0.82 | 0.88 | 0.86 |
| RMSE [ms$^{-1}$] | 0.044 | 0.032 | 0.184 | 0.412 |
| $R^2$, quadratic model | 0.78 | 0.85 | 0.93 | 0.92 |
| RMSE [ms$^{-1}$] | 0.017 | 0.055 | 0.14 | 0.316 |

**Table 3. The percentage of correctly detected vertical wind direction (F) was compared as a ratio of the number of vertical wind directions successfully identified using FODS to the number of chosen data based on the sonic anemometer wind direction. A 5 m length of FODS data 12 to 17 m agl was used to compute the F for both towers. The same data as Fig.7 was used for computation.**

| | Main tower | | Turbulence tower | |
|---|---|---|---|---|
| | N | F (%) | N | F (%) |
| All of the data | 4200 | 60 | 4200 | 63 |
| $\rho_{roll} > 0.8$ | 834 | 71 | 753 | 67 |