# Peer review of "Toward quantifying turbulent vertical airflow and sensible heat flux in tall forest canopies using fiber-optic distributed temperature sensing"

_Atmospheric Measurement Techniques, 2022_

## Referee Comment (RC2)

[referee-annotated manuscript omitted]

---

## Referee Comment (RC3)

**Toward quantifying turbulent vertical airflow and sensible heat flux in tall forest canopies using fiber-optic distributed temperature sensing**

5 Mohammad Abdoli1, Karl Lapo1,2, Johann Schneider1, Johannes Olesch1, Christoph K. Thomas1,2

[referee-annotated manuscript omitted]

---

## Author Comment (AC2)

**Respond to reviewer #2: Q/C (Question/Comment);R(Response)**

Thank you so much for dedicating significant time to read and comment on our manuscript, and the level of scrutiny you exercised. We appreciate your detailed comments and recommendation for the manuscript. Here, in the open discussion period, we would like to address your main concerns and keep responding to all your remaining comments to the final revision.

**Line** 112:

**Q/C:** Why10 min instead of the usual 30 min perturbation time scale?

**R**: According to Vickers et al. (2009) for the sun-lit daytime flux period using a perturbation time scale of 10 min reduces the random sampling error significantly compared to 30 min. Reducing the perturbation time scale from 30 to 10 min increases the systematic error only by a few percent, which is small compared to other sources of uncertainty. While 30 min is often used in the literature, it is often not the optimal choice. It depends on a study's objectives. Our intention was to extract the turbulent flux (as opposed to other non-turbulent contributions to the mass and heat exchange) while limiting its overall uncertainty. To this end, 10 min is an excellent choice.

**Line** 171:

**Q/C**: what is the role of TKE on it?! It should be the result of the convective heat flux and associated with windspeed, shouldn't it?!

**R:** The convective heat loss from coned fibers scales with TKE; please refer to figure 5 in Lapo et al. (2020). TKE by definition, is proportional to the squared turbulent velocity perturbations. Yes, TKE enhances the heat loss from the heated coned fibers and is the reason why a difference in fiber temperatures is sensed.

**Line** 192:

**Q/C**: Why not?! Could you comment on it?!

**R:** Rotation angles in weak-wind environments such as forest canopies are often much larger compared to those above strong-wind short-statured vegetative canopies and may introduce additional uncertainty rather than helping to fulfill the theoretical requirement of Reynold'a second postulate. In the roughness sublayer the flow – by definition – responds to the presence of individual flow obstacles, including tree trunks, stems, and undergrowth, which often leads to a significant wind direction dependence of the rotation angles. Here, we decided not to use any rotation for the

quadrant analysis for those reasons but have not conducted a detailed investigation of the resulting uncertainties. We may do so for the final round of revisions and include the results in the manuscript if important implications arise.

**Line** 251:

**Q/C**: Could you explain why this option was chosen?!

**R:** We increased the length of the shroud from 1.5 m at EBG (Part 1) to 3 m for the forest environment (Part 2) because the minimum resolvable scale with fiber-optic cable and DTS device used in this experiment is 30 s (Freundorfer et al. 2021). In the preparatory phase of the experiment, our analyses yielded a mean magnitude of the vertical wind speed perturbations of 0.1 ms$^{-1}$ for the Waldstein subcanopy site; hence a shroud length of at least 30 s * 0.1 ms$^{-1}$= 3 m seemed optimal to sample the passing eddies by FODS. The turbulence spectrum in rough forest canopies is dominated by organized turbulent motions resulting in more low-frequency turbulence compared to short-vegetated grasslands; hence the integral length scale is larger. This adjustment seemed necessary to capture to main energy-containing eddies.

**Line** 262:

**Q/C**: So, if you found this, what are the reasons to publish the paper/results?! Only to show the idea?! I did not understand the real application. Could you please describe it better?!

**R:** We think there is a misunderstanding here. In the first part of the study, we designed and tested the shroud aimed at observing the vertical flow using FODS in the forest environment; and it failed for the reasons reported. However, we believe reporting this failure has merit as often one learns more from failure than from success. We want to emphasize that failure is always part of scientific work, and the community can also learn from failures. The general fluid dynamics and engineering approach though is physically plausible and could have yielded quantitative results. However, in the second part of the study, we report on a significant finding from an alternative, unshrouded section of the fiber-optic cable, which we tested in comparison. Here, FODS could observe the vertical turbulent flow even without a shroud during the weak wind situation, specifically at the height where the minimum horizontal wind speed in the sub-canopy occurs. This is a significant finding propelling the FODS technique from measuring only first-order to meaningful second-order statistics, including vertical fluxes.

**Line** 352:

**Q/C**: is this conclusion associated with the observations made at **line** 262?

**R:** The reported failure is related to the first part of the study in which the shroud did not yield meaningful results, while the reported conclusion relates to the second part of the paper, where the coned fibers could observe the coherent structures without a shroud. In our revisions, we will clarify this misunderstanding; thanks for bringing this to our attention.

**Line** 540:

**Q/C**: I was expecting to have ejection/sweeps not occurring at the same time,... But at this Figure 6, there are a lot of time interval that both occurs at the same time. See it (as an example), before 5:10 and 5:20 h,

**R:** As you have mentioned, the sweep and ejection phases are not happening simultaneously at any one location, while they do, of course, in a sufficiently larger fluid volume as the sweep-ejection phases are continuous cycles of the mixing-layer scale vortices. The illustration of 1h of high-resolution data leads to believe that they do co-occur at the single measurement location. We will adjust the time axis during final revisions.

**Line** 575:

**Q/C**: why the sensible heat flux is mostly positive during nighttime?! Please, explain it

It occurs due to typical counter-gradient fluxes in the sub-canopy induced by non-colocated heat sources with cooling at the top of the canopy and warming from the forest floor and subcanopy. It is a known phenomenon in forest canopies, see e.g. Denmead & Bradley (1985) .

**Counter-gradient flux definition with AMS Glossary:**

A flux of some variable opposite to the mean gradient of that variable. For example, if temperature decreases upward, then a counter-gradient heat flux would be downward, from cold to hot. While this appears to violate a law of thermodynamics that states heat flows from hot to cold, those laws are found not to be violated when nonlocal motions (air parcels moving across finite distances) are considered. Flux is not caused by, nor related to, the local gradient when coherent structures are present.

**References:**

Vickers, D., Thomas, C., & Law, B. E. (2009). Random and systematic CO2 flux sampling errors for tower measurements over forests in the convective boundary layer. agricultural and forest meteorology, 149(1), 73-83.

Denmead, O. T., & Bradley, E. F. (1985). Flux-gradient relationships in a forest canopy. In The forest-atmosphere interaction (pp. 421-442). Springer, Dordrecht.

---

## Author Comment (AC3)

**Respond to reviewer #3:**

The authors present a very challenging approach to measure wind and sensible heat flux with fiber optic distributed temperature sensing. Unfortunately, the results of the study are not so promising as it appeared that the shroud did not work well to measure vertical wind speed. I appreciate the authors braveness to submit (partly) failed experiments. Also failed experiments can help the community to learn. Having said this, I think the current manuscript needs major revisions as it seems more a good draft then a full paper (yet). Problem statement and method section miss essential information and certain choices are not well explained. Additionally, I think that also many of the figures can be improved with less abbreviations and make them more self-explanatory (e.g., by giving the figure titles as 'setup 1, setup 2'. this would also reduce the caption lenght). In the attached pdf I commented in detail. Here I only indicate my main comments.

We appreciate your time and effort in reading through the manuscript. We found your comments very helpful and will revise our manuscript in response. Here in the open discussion period, we respond to your general questions and main comments. We will address all remaining detailed comments during the subsequent revisions. Based on your comments, we will add more details to the problem statement and the method section and revise the figures to make them clearer and more readable.

1- The outline of the study is that the authors first investigate several shroud configurations on a grass field (EBG). The 'best' shroud is then later used in a follow-up experiment in a forest. However, in the method section there is barely any information on the different setups and why shroud color, mesh size, rigidity or shape would affect the measurements. What were the design criteria. This part should be extended and improved.

   We will update the details of the shroud design and criteria in the manuscript. About the shroud design, we iterated multiple should configurations in different diameters (1 and 0.6 m), gray and white colors, small and large pore size should, and a shroud with and without supporting metal mesh underneath the shroud. The reasoning line for each selection is:

   I.   Diameter: The task was to design a shroud to eliminate the horizontal flow while keeping the vertical flow perturbation intact. We thought increasing the shroud diameter could increase the horizontal flow disturbances inside the shroud since it offers a larger pathway for airflow. On the other hand, decreasing the shroud diameter and

placing it close to the sonic anemometer could cause systematic turbulence created with the shroud itself.

II. Length: We determined the length of the shroud over the grass to be long enough to accommodate the typical length scales of the vertical turbulent flow, keep the sensors away from shroud structure-induced flow disturbances, and be feasible to install, given the available hardware and facilities.

III. Color: We used the gray shroud first. Initial results showed substantial heating of the shroud material during daytime conditions inducing strong upward directed (free-) convective heat transfer and thus distorting flow statistics inside the shroud. In response, we changed the shroud's color to white to avoid possible errors, together with increasing the pore size of the shroud,

IV. Mesh size: The initial mesh size was selected based on the previous experiments in the group and then improved based on the initial results.

V. Rigidity: The very first setup of the shroud was designed without supporting mesh and was just a tensioned shroud with two rings at the top and bottom. We observed that the shroud gets very unstable during wind gusts and induces uninvited turbulence. We decided to make the shroud rigid enough to avoid this problem.

2- Why are not all the shroud experiments (EBG) compared to the sonic (thus also setup 1 and 2)? Now the benchmark is the 'unshrouded' FODS measurements, which is also an experimental method. I would benchmarkt the shrouds to the sonic as this is likely closer to the truth.

We could not understand this comment completely. All the conducted statistics in the EBG experiment and the forest are benchmarked to the sonic observations. If the question means bringing FODS in the x-axis in figures 7 and 8, we will swap the axis to plot the FODS against sonic. Otherwise, we would appreciate it if you clarify the comment.

3- Base the first test, the authors pick 'the best shroud' setup to apply it in a forest. Only surprising change, is that 'suddenly' the shroud lenght is increased. While from study 1 the authors could have learnd that dimensions matter for the wind direction. This is in my view a major shortcoming of this paper.

This question was also raised by reviewer #2. We provide the same answer here as well.

We increased the length of the shroud from 1.5 m at EBG (Part 1) to 3 m for the forest environment (Part 2) because the minimum resolvable scale with fiber-optic cable and DTS device used in this experiment is 30 s (Freundorfer et al. 2021). In the preparatory phase of the experiment, our analyses yielded a mean magnitude of the vertical wind speed perturbations of 0.1 ms$^{-1}$ for the Waldstein subcanopy site; hence a shroud length of at least 30 s * 0.1 ms$^{-1}$= 3 m seemed optimal to sample the passing eddies by FODS. The turbulence spectrum in rough forest canopies is dominated by organized turbulent motions resulting in more low-frequency turbulence compared to short-vegetated grasslands; hence the integral length scale is larger. This adjustment seemed necessary to capture to main energy-containing eddies.

4- Despite the admitted 'failure' of the forest experiments, the authors still show the initial plan to calculate the sensible heat flux. But what is the value of this, once the wind speed measurements are not correct?
We believe there is a misunderstanding. We computed the distributed sensible heat flux using FODS for all heights (0 to 30 m agl), but chose to report the sensible heat flux for the unshrouded part (12 to 17 m agl) to compare against the eddy-covariance estimates because the vertical wind component at this height range shows the most promising signal to noise ratios. In other words, we compute the sensible heat flux based on the successful part of the FODS section. Our experimental design did not yield meaningful results for the shroud heights ranging from 2.5 to 5.5 m agl (see Fig. 2).

5- The reference list contains 34 references, from which 16 are from the own research group. This is almost 50%! I highly recommend to put the study into a more broad context. Many other groups also worked in this study field, including groups that also work with FODS
Thank you for bringing this issue to our attention. We will revise the references to make them less group-centric, particularly for the general forest turbulence sections. However, the FODS community is still small, and most researchers are interrelated and learned the FODS technique from our work group. Many author names you suggested are coauthors of our group members in the cited literature. We look forward to more researchers discovering and applying the utility of FODS techniques in the future.

In the attachment, I added some suggestions.

We will address all of your detailed comments for the revised version. Thank you again for your helpful comments and recommendation.

---

## Author Response (AR1)

Dear Prof. Cléo Quaresma Dias-Junior,

We have included the reviewer comments and responded to them individually, indicating exactly how we addressed each concern or problem and describing the changes we have made. We added more detailes and reasonings on method section and also clarified the figures to make them more readable.

We hope the revised manuscript will better suit the Journal but are happy to consider further revisions, and we thank you for your time and considerations.

Sincerely yours,

Paper authors

**Respond to reviewer #2: Q/C (Question/Comment);R(Response)**

Thank you so much for dedicating significant time to read and comment on our manuscript, and the level of scrutiny you exercised. We appreciate your detailed comments and recommendation for the manuscript.

**Line** 12:

**Q/C**: if is vertical, then you should say only component (not winspeed /direction)

**R:** Revised accordingly for entire manuscript.

**Line** 15: Too long for an abstract

**R:** Shortened

**Line** 35:

**Q/C**: I guess that you can write this at conclusions, but not at the abstract

**R:** Moved to conclusion

**Line** 55, 60, 63:

**Q/C**: the correct reference is Thomas et al., 2012

R: The citation style changed to copernicus format and all of the relating problems solved.

**Line** 68 :

**Q/C**: Do these WT experiments had thermal gradient included or only neutral conditions?!

**R:** The wind tunnel experiment is carried out at ambient temperature without a thermal gradient.

**Line** 75:

**Q/C**: Highlighted

R: Corrected

**Line** 93, 94:

**Q/C**: Highlighted

R: Revised

**Line** 112:

**Q/C**: Why10 min instead of the usual 30 min perturbation time scale?

R: According to Vickers et al. (2009) for the sun-lit daytime flux period using a perturbation time scale of 10 min reduces the random sampling error significantly compared to 30 min. Reducing the perturbation time scale from 30 to 10 min increases the systematic error only by a few percent, which is small compared to other sources of uncertainty. While 30 min is often used in the literature, it is often not the optimal choice. It depends on a study's objectives. Our intention was to extract the turbulent flux (as opposed to other non-turbulent contributions to the mass and heat exchange) while limiting its overall uncertainty. To this end, 10 min is an excellent choice.

**Line** 116:

**Q/C**: Could you present a synoptic analysis during the field work (Oct-Nov 2020)?!

R: Added to manuscript section 2.2.

**Line** 118:

**Q/C**: Highlighted

R: Revised

**Line** 120:

**Q/C**: italic as it is latin names...

R: Revised

**Line** 137:

**Q/C**: in the first part (test) you have used 20 Hz, but not it is 10 Hz. Could you explain why the difference?! What about the time interval, is it 10 or 30 min?!

This statement "It should be noted that all of the eddy covariance data in this stude were sampled at 20Hz, with the exception of the permanent eddy covariance station at the turbulence tower, which samples at 10Hz. All of the eddy covariance systems in this study use the same eddy covariance data processing and flux computatoion routine." added to the end of section 2.2

**Line** 188:

**Q/C**: 20 Hz was used in the test and 10 Hz was used in the field experiment. Which one?

R: This eddy covariance system is the long-term eddy covariance system at the Waldstein site and samples with 10Hz rather than other temporarily installed eddy covariance systems, which all sampled with 20Hz.

**Line** 148, 149:

**Q/C**: K or k???

R: K is correct.

**Line** 171:

**Q/C**: what is the role of TKE on it?! It should be the result of the convective heat flux and associated with windspeed, shouldn't it?!

R: The convective heat loss from coned fibers scales with TKE; please refer to figure 5 in Lapo et al. (2020). TKE by definition, is proportional to the squared turbulent velocity perturbations. Yes, TKE enhances the heat loss from the heated coned fibers and is the reason why a difference in fiber temperatures is sensed.

**Line** 175:

**Q/C**: a or b? Check it and correct in the text

R: Corrected

**Line** 182, 188:

**Q/C**: Highlighted

R: Corrected

**Line** 185, 186

**Q/C**: Highlighted

R: Revised

**Line** 192:

**Q/C**: Why not?! Could you comment on it?!

R: 3D coordinate rotation applied to the data and the results updated accordingly.

**Line** 194:

**Q/C**: I wonder that this classification has been studied (and defined) before so it should be cited...

R: Citation added

**Line** 206 :

**Q/C**: This has been written before. I suggest to delete it to reduce the lenght of the manuscript

R: Done!

**Line** 214:

**Q/C**: Highlighted

R: corrected

**Line** 216:

**Q/C**: repetead ...

R: Eliminated

**Line** 220:

**Q/C**: do you believe that your measurements have this accurancy?! 3 digits?!

R: The number of digits revised for the entire manuscript.

**Line** 223: this is wind component, isn't it?!

R: Yes, we renamed it as total wind speed by adding the equation to the manuscript.

**Line** 224, 225:

**Q/C**: Highlighted

R: Revised

**Line** 234:

**Q/C**: based of followed?

R: Revised

**Line** 237:

**Q/C**: Highlighted

R: Revised

**Line** 241:

**Q/C**: s - please correct it at all manuscript

R: Revised

**Line** 245:

**Q/C**: This is more conclusions than discussion

R: Merged with conclusion

**Line** 251:

**Q/C**: Could you explain why this option was chosen?!

R: The following discussion added to the manuscript:

We increased the length of the shroud from 1.5 m at EBG (Part 1) to 3 m for the forest environment (Part 2) because the minimum resolvable scale with fiber-optic cable and DTS device used in this experiment is 30 s (Freundorfer et al. 2021). In the preparatory phase of the experiment, our analyses yielded a mean magnitude of the vertical wind speed perturbations of 0.1 ms$^{-1}$ for the Waldstein subcanopy site; hence a shroud length of at least 30 s * 0.1 ms$^{-1}$ = 3 m seemed optimal to sample the passing eddies by FODS. The turbulence spectrum in rough forest canopies is dominated by organized turbulent motions resulting in more low-frequency turbulence compared

to short-vegetated grasslands; hence the integral length scale is larger. This adjustment seemed necessary to capture to main energy-containing eddies.

**Line** 255:

**Q/C**: may be "once " is better here...

R: Corrected

**Line** 256:

**Q/C**: If you are comparing values, they should have the same significant digits... Also, 0.024 is almost null that means independent... Could the authors comment that?!

R: Revised

**Line** 262:

**Q/C**: So, if you found this, what are the reasons to publish the paper/results?! Only to show the idea?! I did not understand the real application. Could you please describe it better?!

R: We think there is a misunderstanding here. In the first part of the study, we designed and tested the shroud aimed at observing the vertical flow using FODS in the forest environment; and it failed for the reasons reported. However, we believe reporting this failure has merit as often one learns more from failure than from success. We want to emphasize that failure is always part of scientific work, and the community can also learn from failures. The general fluid dynamics and engineering approach though is physically plausible and could have yielded quantitative results. However, in the second part of the study, we report on a significant finding from an alternative, unshrouded section of the fiber-optic cable, which we tested in comparison. Here, FODS could observe the vertical turbulent flow even without a shroud during the weak wind situation, specifically at the height where the minimum horizontal wind speed in the sub-canopy occurs. This is a significant finding propelling the FODS technique from measuring only first-order to meaningful second-order statistics, including vertical fluxes.

**Line** 270:

**Q/C**: quantify it, please

R: The statement updated to include the contribution percentage of coherent structure to sensible heat flux at each subcanopy level.

**Line** 281:

**Q/C**: Highlighted

R: Done

**Line** 291:

**Q/C**: Highlighted

R: Revised

**Line** 313:

**Q/C**: this is almost independent...

R: Rephrased

**Line** 322:

**Q/C**: please quantify and explain those behaviours

R: Quantitative errors added to the manuscript

**Line** 325:

**Q/C**: Highlighted

R: Done

**Line** 339, 340,34, 349:

**Q/C**: highlighted

R: Done

**Line** 343:

**Q/C**: this is true, but this results is very old. For tropical forest, it has been written in the 80s

R: Eliminated

**Line** 351:

**Q/C**: is this conclusion associated with the observations made at **Line** 262?

R: No, this result is based on the FODS part outside of the shroud, which yields promising results.

**Line** 371:

**Q/C**: Highlighted

R: Corrected

**Line** 388:

**Q/C**: Highlighted

R: Corrected

**Line** 395:

**Q/C**: same reference

R: Corrected

**Line** 501, 525, 531:

**Q/C**: Highlighted

R: Revised

**Line** 539:

**Q/C**: I was expecting to have ejection/sweeps not occurring at the same time,... But at this Figure 6, there are a lot of time interval that both occurs at the same time. See it (as example), before 5:10 and 5:20 h,

R: As you have mentioned, the sweep and ejection phases are not happening simultaneously at any one location, while they do, of course, in a sufficiently larger fluid volume as the sweep-ejection phases are continuous cycles of the mixing-layer scale vortices. The illustration of 1h of high-resolution data leads to believe that they do co-occur at the single measurement location.

We added another plot by zooming into the 10 min range to make it clear.

**Line**  546:

**Q/C**: or 10 Hz?

R: 20 Hz in subcanopy and 10 Hz in turbulence tower

**Line** 552:

**Q/C**: Highlighted

R: Revised for the entire manuscrip

**Line** 575:

**Q/C**: why the sensible heat flux is mostly positive during nighttime?! Please, explain it

R: It occurs due to typical counter-gradient fluxes in the sub-canopy induced by non-colocated heat sources with cooling at the top of the canopy and warming from the forest floor and subcanopy. It is a known phenomenon in forest canopies, see e.g. Denmead & Bradley (1985) .

Counter-gradient flux definition with AMS Glossary:

A flux of some variable opposite to the mean gradient of that variable. For example, if temperature decreases upward, then a counter-gradient heat flux would be downward, from cold to hot. While this appears to violate a law of thermodynamics that states heat flows from hot to cold, those laws are found not to be violated when nonlocal motions (air parcels moving across finite distances) are considered. Flux is not caused by, nor related to, the local gradient when coherent structures are present.

**References:**

Vickers, D., Thomas, C., & Law, B. E. (2009). Random and systematic CO2 flux sampling errors for tower measurements over forests in the convective boundary layer. agricultural and forest meteorology, 149(1), 73-83.

Denmead, O. T., & Bradley, E. F. (1985). Flux-gradient relationships in a forest canopy. In The forest-atmosphere interaction (pp. 421-442). Springer, Dordrecht.

**Respond to reviewer #3:**

**General comments:**

The authors present a very challenging approach to measure wind and sensible heat flux with fiber optic distributed temperature sensing. Unfortunately, the results of the study are not so promising as it appeared that the shroud did not work well to measure vertical wind speed. I appreciate the authors braveness to submit (partly) failed experiments. Also failed experiments can help the community to learn. Having said this, I think the current manuscript needs major revisions as it seems more a good draft then a full paper (yet). (I) **Problem statement and method section miss essential information and certain choices are not well explained**. Additionally, I think that also (II) **many of the figures can be improved with less abbreviations and make them more self-explanatory** (e.g., by giving the figure titles as 'setup 1, setup 2'. this would also reduce the caption lenght). In the attached pdf I commented in detail. Here I only indicate my main comments.

We appreciate your time and effort in reading through the manuscript. We found your comments very helpful and will revise our manuscript in response. Here is our response to the two abovementioned comments:

I - We added more details to the method section explaining the shroud specifications and the reasoning behind each choice.

II - We adjusted the figures in the manuscript to be self-explanatory, taking into account both reviewer's detailed comments.

1- The outline of the study is that the authors first investigate several shroud configurations on a grass field (EBG). The 'best' shroud is then later used in a follow-up experiment in a forest. However, in the method section there is barely any information on the different setups and why shroud color, mesh size, rigidity or shape would affect the measurements. What were the design criteria. This part should be extended and improved.

We updated the details of the shroud design and criteria in the manuscript. About the shroud design, we iterated multiple should configurations in different diameters (1 and 0.6 m), gray and white colors, small and large pore size should, and a shroud with and without supporting metal mesh underneath the shroud. The reasoning line for each selection is:

I.    Diameter: The task was to design a shroud to eliminate the horizontal flow while keeping the vertical flow perturbation intact. We thought

increasing the shroud diameter could increase the horizontal flow disturbances inside the shroud since it offers a larger pathway for airflow. On the other hand, decreasing the shroud diameter and placing it close to the sonic anemometer could cause systematic turbulence created with the shroud itself.

II.   Length: We determined the length of the shroud over the grass to be long enough to accommodate the typical length scales of the vertical turbulent flow, keep the sensors away from shroud structure-induced flow disturbances, and be feasible to install, given the available hardware and facilities.

III.   Color: We used the gray shroud first. Initial results showed substantial heating of the shroud material during daytime conditions inducing strong upward directed (free-) convective heat transfer and thus distorting flow statistics inside the shroud. In response, we changed the shroud's color to white to avoid possible errors, together with increasing the pore size of the shroud,

IV.   Mesh size: The initial mesh size was selected based on the previous experiments in the group and then improved based on the initial results.

V.   Rigidity: The very first setup of the shroud was designed without supporting mesh and was just a tensioned shroud with two rings at the top and bottom. We observed that the shroud gets very unstable during wind gusts and induces uninvited turbulence. We decided to make the shroud rigid enough to avoid this problem.

2- Why are not all the shroud experiments (EBG) compared to the sonic (thus also setup 1 and 2)? Now the benchmark is the 'unshrouded' FODS measurements, which is also an experimental method. I would benchmarkt the shrouds to the sonic as this is likely closer to the truth.

This question was unclear to us. We compare the shrouded setup against the sonic anemometer at EBG. In the forest, the FODS is also compared to the sonic at 4m at the main tower and 36m at the turbulence tower.

3- Base the first test, the authors pick 'the best shroud' setup to apply it in a forest. Only surprising change, is that 'suddenly' the shroud lenght is increased. While from study 1 the authors could have learnd that dimensions matter for the wind direction. This is in my view a major shortcoming of this paper.

We included the following reasoning in the manuscript:

We increased the length of the shroud from 1.5 m at EBG (Part 1) to 3 m for the forest environment (Part 2) because the minimum resolvable scale with fiber-optic cable and DTS device used in this experiment is 30 s (Freundorfer et al. 2021). In the preparatory phase of the experiment, our analyses yielded a mean magnitude of the vertical wind speed perturbations of 0.1 ms$^{-1}$ for the Waldstein subcanopy site; hence a shroud length of at least 30 s * 0.1 ms$^{-1}$= 3 m seemed optimal to sample the passing eddies by FODS. The turbulence spectrum in rough forest canopies is dominated by organized turbulent motions resulting in more low-frequency turbulence compared to short-vegetated grasslands; hence the integral length scale is larger. This adjustment seemed necessary to capture to main energy-containing eddies.

4- Despite the admitted 'failure' of the forest experiments, the authors still show the initial plan to calculate the sensible heat flux. But what is the value of this, once the wind speed measurements are not correct?
We believe there is a misunderstanding. We computed the distributed sensible heat flux using FODS for all heights (0 to 30 m agl), but chose to report the sensible heat flux for the unshrouded part (12 to 17 m agl) to compare against the eddy-covariance estimates because the vertical wind component at this height range shows the most promising signal to noise ratios. In other words, we compute the sensible heat flux based on the successful part of the FODS section. Our experimental design did not yield meaningful results for the shroud heights ranging from 2.75 to 5.75 m agl (see Fig. 2).

5- The reference list contains 34 references, from which 16 are from the own research group. This is almost 50%! I highly recommend to put the study into a more broad context. Many other groups also worked in this study field, including groups that also work with FODS
Thank you for bringing this issue to our attention. We will revise the references to make them less group-centric, particularly for the general forest turbulence sections. However, the FODS community is still small, and most researchers are interrelated and learned the FODS technique from our work group. Many author names you suggested are coauthors of our group members in the cited literature. We look forward to more researchers discovering and applying the utility of FODS techniques in the future.

**Detailed comments:**

**Line 98:**

**Q/C**: I think more information on the shroud is needed. What were the design considerations? Why would color matter, why would shroud diameter matter? Please explain and elaborate

R: done!

**Line 127:**

**Q/C**: 2,75m right?

R: Corrected

**Line 141**

**Q/C**: why is this needed? Please explain in relation to your work.

R: Explanation statement added to the manuscript.

**Line 160**

**Q/C**: Wm-1 (capital)

R: Corrected!

**Line 186**

**Q/C**: Define positive direction of u and w

R: Defined!

**Line 212**

**Q/C**: why are the wind speeds not compared to the sonic? Wouldn't that be more fair comparison?

R: The comparison result of wind speed is included in Table 1. Nevertheless, since we are interested in how the vertical turbulence varies inside the shroud, we decided to present the scatter plot of $\sigma_w$.

**Line 218**

**Q/C**: this can also be due to the higher N

R: We reduced the length of data used in setup 3 to N = 552 to make it consistent with the other two setups.

**Line 229**

**Q/C**: italic

R: Corrected

**Line 235**

**Q/C**: font size seems to be smaller

R: Corrected

**Line 251**

**Q/C**: did you investigated is this influence the windspeed measurements?

R: We investigate the method in subcanopy and under weak wind conditions (e.g., wind speed $< 0.2$ ms$^{-1}$). In this situation in which the horizontal wind is really weak, increasing the shroud length from 1.5m to 3m may not affect the wind flow inside the shroud enormously. However, in the case of applying this method to a higher range of wind speeds, the effect of shroud length may become important.

**Line 256**

**Q/C**: ah... correlation coefficient dT and w.... maybe this can be clarified in the text

R:Double checked that the $\rho_w$ and $\rho_{roll}$ are defined in the section 3.2 clearly. The definition also added to the Table 2.

**Line 263**

**Q/C**: I already found it weird to suddenly increase the height of the shroud... the problem with this fact is that the EBG and Waldstein forest study are now disconnected (while they are in one paper)

R: We included the reasonings to the manuscript (also included here in response to reviewer #3, general comments, No. 3 )

**Line 310**

**Q/C**: I missed in the objective that the aim was to estimate H purely based on FODS.

R: The statement added to the research objective at the end of introduction.

**Line 457**

**Q/C**: ±50% of the references are from the own research group. It is highly recommended to look for relevant others studies (e.g., work by Yu Cheng, Bart Schilperoort, Robert Predosa, Matth Zeeman).

R: As responded in general comments, the references ….

**Line 475, Fig. 1**

**Q/C**: annotate this figures, with where the instruments are located. Also add the installation heights of the instruments

R: Done!

**Line 495, Fig. 2**

**Q/C**: what is this distance? (distance tower and dts setup), added

I think the left coned fibre should be colored blue as it is non-heated

R: Both coned fibers are heated

Not sure is this correctly depicted? Corrected

not sure if this is a common abbreviation? Corrected

**Line 525 , Fig. 4**

**Q/C**: be consistent in your names. In the method section you do not describe setup 1, 2 and 5. Which one is which setup?

R: Fig 4 updated with new labels and also fewer data points for setup 3.

**Line 530, Fig. 5**

**Q/C**: clarify meaning of Ts, and v (only w and u are defined)

chp?

R: chp stands for "change point" and Ts stands for sonic temperature , added to the manuscript.

**Line 565, Fig. 9, Fig. 10**

**Q/C**: why did you compare more at the same height? 12-17 is likely in the canopy while the sonic is above the canopy?

R: The fiber-optic array at 4 m is inside of the shroud which does not yield a good results. The most promising signals of vertical ait movement that we get from fiber-optic is at 12-17 m. That's why we chose this height to compare the range of computed fluxes.

Remove the "First-ever"

R: Done

**Line 591, Table 1**

**Q/C**: why are there almost 3x more experiments for setup 3? This might also influence the error metrics.

R: The data of setup 3 shortened to the length of other two experiments, and the computations updated accordingly.